# Replacing Endoscopy with Magnetic Resonance Enterography for Mucosal Activity Assessment in Terminal Ileal Crohn’s Disease: Are We There Yet?

**DOI:** 10.3390/diagnostics13061061

**Published:** 2023-03-10

**Authors:** Anuj Bohra, Abhinav Vasudevan, Numan Kutaiba, Daniel Ross Van Langenberg

**Affiliations:** 1Department of Gastroenterology, Eastern Health, Box Hill, Melbourne 3128, Australia; 2Department of Radiology, Eastern Health, Box Hill, Melbourne 3128, Australia

**Keywords:** magnetic resonance enterography, Crohn’s disease, diagnostics, endoscopy

## Abstract

Crohn’s disease (CD) is a chronic immune mediated disorder that most commonly affects the small bowel and/or the large bowel. Treatment targets in CD include mucosal healing assessed via ileocolonoscopy and transmural healing assessed through cross-sectional imaging modalities such as magnetic resonance enterography (MRE). More recently, histological healing in CD has emerged as a treatment target, though it is made cumbersome given its reliance on frequent endoscopic examinations. With expert guidelines now recommending regular objective assessments as part of a treat-to-target approach, accurate non-invasive assessment will become increasingly critical. MRE has an established role in the assessment of small bowel CD, with growing data supportive of its ability in detecting disease activity at mucosal and histological levels. This could therefore potentially reduce the need for serial endoscopic assessment. Thus, this review will assess the capacity of individual MRE parameters and MRE indices for detecting mucosal and histological small bowel CD activity. Furthermore, challenging scenarios, such as CD activity detection in post-operative clinical scenarios and abnormal findings in the context of a normal ileocolonoscopy, will be explored.

## 1. Introduction

Crohn’s Disease (CD) is a chronic inflammatory condition with an incident rate of over 1 in 300 in developed countries [1]. Though CD can affect the entire gastrointestinal tract, approximately 70% of patients will have small intestinal involvement, with endoscopic manifestations including ulceration and inflammation, which can progress to penetrating and/or stricturing complications when untreated [2,3,4,5]. Advances in medical therapy and the availability of multiple different classes of agents have culminated in more stringent treatment targets plus the need for regular objective assessments of disease activity to ensure optimal and durable outcomes for patients.

The gold standard for CD activity assessment remains ileocolonoscopic examination [6], with established definitions for endoscopic remission such as a Simplified Endoscopy Score in CD (SES-CD) of ≤2 points or Crohn’s Disease Endoscopic Index of Severity (CDEIS) <3 and a lack of ulceration [7,8]. However, ileocolonoscopy has several limitations that preclude regular and repeated endoscopic assessment, including cost and resource constraints, the need for bowel preparation and sedation anaesthesia, plus procedural risks including bowel perforation and bleeding. Additionally, assessment of the small bowel is limited to the terminal ileal mucosa with a standard ileocolonoscopy, yet the proximal small bowel and deeper submucosal layers cannot be evaluated [9].

Recent advances in the quality and accuracy of non-invasive imaging modalities have resulted in these measures being preferentially used as surrogates for endoscopic assessment, and guidelines support the use of magnetic resonance enterography (MRE), intestinal ultrasound (IUS), and computed tomography enterography (CTE) as adjuncts to ileocolonoscopy [7]. The key goal of any non-invasive method of CD activity detection is to achieve equivalent sensitivity to that of an ileocolonoscopy, including mucosal activity. Indeed, the advantages of non-invasive MRE, including the provision of transmural and extra-mural information, convenience, and improving access and cost versus the known disadvantages of endoscopy, raise the question as to whether MRE could even supersede ileocolonoscopy as the first-line investigation in small bowel CD. Hence, this review aims to evaluate the ability of MRE imaging to detect small bowel CD activity, with a particular focus on imaging parameters and their respective correlation with endoscopic and histological activity assessments. Other aspects pertaining to the technical performance of MRE relevant to the detection of endoscopic and mucosal activity in CD will also be discussed.

## 2. MRE Parameters That Correlate with Disease Activity

Parameters in cross-sectional imaging are not specific to CD, and interpretation must account for the individual’s clinical context. Additionally, inflammatory and fibrostenosing CD can produce convergent features on MRE(see Table 1). Multiple luminal and extra-luminal MRE parameters of disease activity are described and will be discussed, with a particular focus on the individual parameter’s ability to detect mucosal activity and ulceration that would typically be achieved with endoscopic assessment.

### 2.1. Bowel Wall Thickness

Bowel wall thickness (BWT) is a shared parameter across the cross-sectional imaging modalities and is validated as a marker of CD activity (see Figure 1, Figure 2 and Figure 3) [10]. Recent consensus statements depict a BWT > 3 mm as abnormal in either the terminal ileum or colon [11,12]. As an individual parameter, BWT has been shown to be an independent predictor of active mucosal disease; however, increased BWT may also reflect fibrostenosing CD [13,14]. Within mucosally active CD, multiple studies have shown the presence of ulcers endoscopically to be a critical marker of severity, with ulcerated segments more likely to demonstrate increased overall BWT on MRE [10,15,16,17]. In a prospective study of 48 patients with CD undergoing paired MRE and colonoscopy for disease activity assessment, a significant difference in BWT between non-ulcerated and ulcerated endoscopic segments (4.29 ± 1.46 mm vs. 5.61 ± 1.26 mm, *p* < 0.001) was observed. Moreover, a further significant reduction in mean BWT was seen in endoscopically normal mucosa and non-ulcerated inflammation (2.83 ± 0.61 mm vs. 4.29 ± 1.46 mm, *p* < 0.001) [18]. In the absence of ulceration on endoscopy, increased BWT as a parameter to assess CD activity appears to be less sensitive and specific; thus, additional parameters should be considered. In two subsequent studies by the same authors, the prevalence of a BWT >3 mm was only seen in 46 to 74% of patients with mildly active CD on endoscopy [16,19]. Other studies have reported similar findings, with a BWT >3 mm having a sensitivity of only 35% in detecting mucosally active CD in the absence of ulcers or erosions on endoscopic examination [20]. Finally, as an individual parameter, there is good reliability in detecting increased BWT when assessed by experienced radiologists, with Jairath et al. reporting an inter-class correlation of 0.72 (95% confidence interval (CI), (0.62–0.80)) when assessing BWT within the terminal ileum [21].

### 2.2. Bowel Wall Contrast Enhancement

Bowel wall contrast enhancement can aid the differentiation of active luminal CD from fibrostenosing changes. Key findings in post-contrast MRE sequences include the degree and pattern of bowel wall enhancement [11,12]. Patterns of bowel wall enhancement are categorised as homogenous, mucosal, or layered stratification (see Figure 1, Figure 3 and Figure 4). Mucosal pattern enhancement refers to the increased enhancement of the mucosal layer associated with increased BWT. Layered enhancement is defined as the enhancement of the mucosa (inner layer) and serosa (outer layer) with sparing of the middle layer (representing submucosa and muscular layer) [22]. Multiple studies have demonstrated that both mucosal and layered enhancement patterns are associated with histologically active disease [22,23,24,25]. For instance, in a study of 53 CD patients undergoing bowel resection, a preoperative MRE assessment of CD activity demonstrated that both the degree of bowel wall enhancement (mild vs. marked) and the pattern of enhancement (layered vs. homogenous) on delayed T1 post-contrast sequences were altered significantly between different histopathological grades of inflammation assessed on surgical resection specimens [13]. In a subsequent study using ileocolonoscopy as a reference standard, significant differences in bowel wall contrast enhancement were observed between normal mucosa and mucosally active CD [18].

### 2.3. T2 Mural Signal Intensity/Intramural Oedema

T2 mural signal intensity, also known as intramural oedema, is a finding unique to MRE and is not seen with other forms of cross-sectional imaging such as CTE. It is demonstrated by a high signal intensity within the submucosa of bowel wall in affected segments and is best appreciated on T2-weighted fat-suppressed images (see Figure 1 and Figure 3) [11]. Fat suppression allows for differentiation with chronic fibrotic CD, which also presents with a high signal on submucosal T2-weighted images [11]. Intramural oedema is known to correlate with degrees of active inflammation in CD seen via endoscopic assessment and on histology of surgical resection specimens [10,26]. In the absence of endoscopically active CD, oedema is seldom present [16,18]. In a study of 50 patients undergoing paired MRE and colonoscopy, intramural oedema was present in only 5/198 (2.5%) of endoscopically normal bowel segments [16]. Studies have demonstrated that mural signal intensity is able to differentiate grades of endoscopic disease when small bowel ulceration is present; however, it is less sensitive in non-ulcerative, endoscopically active CD [10,18]. In active, ulcerative CD, its prevalence is significantly higher, with studies reporting a sensitivity ranging between 68–84% [16,19,20]. Moreover, intramural oedema demonstrates good reliability, with one study demonstrating an inter-class correlation of 0.80 (95% CI, (0.69–0.89)) within the terminal ileum, with lower reliability in colonic segments [21].

### 2.4. Ulceration

The finding of ulceration on MRE is known to be correlated with active inflammation seen in CD through endoscopic examination and histology [10,26]. On MRE, superficial and deep ulcers can be identified on T2 and post-contrast T1 imaging as areas of deep depressions in the mucosal surface of a thickened segment (see Figure 1) [10,11]. With advanced CD, a cobblestone appearance on MRE can represent areas of deep ulceration alternating with thickened mucosal folds [27]. As expected, in the absence of endoscopically active CD, ulceration is seldom appreciable on MRE, with one study demonstrating a 1.5% false positive rate [16]. Studies have demonstrated the capacity of ulceration on MRE to differentiate between grades of endoscopic CD though its prevalence in MRE, with non-ulcerative endoscopically active CD being reportedly as low as 9% [10]. Endoscopically active CD with visible ulceration is more likely to portend ulceration on MRE, with studies reporting a sensitivity of 53–78% [16,19,20]. Ulceration on MRE has a fair reliability, with one study demonstrating an interclass correlation coefficient of 0.60 (0.45–0.72) within the terminal ileum but reduced reliability in colonic segments [21].

### 2.5. Diffusion-Weighted Imaging

Diffusion-weighted imaging (DWI) is a specialised sequence in MRE. DWI reflects water motility (diffusion) within tissue, with pathological alterations resulting in signal changes (i.e., restricted diffusion) (see Figure 1) [28]. DWI is performed by utilising T2-weighted fat-suppressed sequences and adding a diffusion gradient. The gradient is quantified by a coefficient (b-value) that is used to construct a corresponding apparent diffusion coefficient (ADC) map. A meta-analysis demonstrated that DWI-MRE had a pooled sensitivity and specificity of 93% and 91% in detecting small bowel CD compared to a reference standard including ileocolonoscopy, histopathology, and alternative radiological methods [29]. However, a lack of blinding to the reference standard may reflect bias in these studies and an overestimation of accuracy [29]. When blinded to the reference standard, DWI-MRE sensitivity and specificity in detecting CD was reduced to 79% and 61%, respectively [29]. Qualitative and quantitative assessment of DWI and ADC map signals have been described in the literature. However, the heterogeneity of imaging techniques and studies have limited the use of ADC map thresholds for the detection of active inflammation and the distinction between inflammation and fibrosis. [28].

### 2.6. Stenosis and Pre-Stenotic Dilatation

CD-related stenosis is represented cross-sectionally by a segment of luminal narrowing of 50% compared to the adjacent bowel loop, and in the case of a definite stricture, it is further confirmed by pre-stenotic luminal dilatation (≥3 cm). A ‘probable stricture’ is defined by a luminal narrowing of 50% compared to the adjacent bowel loop without pre-stenotic dilatation (<3 cm) [11]. Specific MRE findings suggestive of an active inflammatory stricture include bowel wall contrast enhancement, BWT, and bowel wall oedema in the affected segment of stenosis [30]. These changes are best appreciated on post-contrast T1 and fat-suppressed T2 image sequences. In addition, the assessment of small intestine motility on cine sequences allows for a differentiation between fixed stenotic lesions and temporary luminal narrowing. When cine sequences are not available, reviewing the site of a possible stricture on different sequences may also provide information as to whether this site remains narrowed or not, given the different sequences are obtained over different timepoints. Regarding fibrotic stenosis/strictures, these are thin-walled and appear hypointense on T1 and T2 sequences and often occur in conjunction with submucosal fibrofatty proliferation. However, as previously mentioned, overlap between inflammatory and fibrotic appearances of strictures is frequently encountered; thus, it is often not possible to differentiate between the two processes [31]. Initial studies demonstrated that MRE had a poor sensitivity in detecting stenosis compared to balloon enteroscopy in small bowel CD, with a reported sensitivity of 40.8% (95% CI, 0.31–0.49) [32]. However, when compared to surgical histology, a recent systematic review reported that MRE detected CD-related strictures with a sensitivity between 75% and 100% and a specificity between 91% and 96% [33].

### 2.7. Motility Sequence Assessment/Cine Sequences

Reduced intestinal motility is well described in patients with active CD, with cine sequences allowing for the assessment of intestinal motility [34]. More recently, an automated software algorithm was developed, facilitating the calculation of the motility index (MI) [35]. Menys et al. demonstrated that in patients with active CD, a reduction in MI was negatively correlated with histologically (*r* = −0.61; (0.7, −0.5)) and endoscopically active CD *(r* = −0.59; (0.7, −0.4)) [36]. In a subsequent study by Dreja et al., the MI was also inversely correlated with mural thickness in the terminal ileum, with a lower MI in patients with active CD versus controls in the ileum [37]. Hence, intestinal motility not only appears to be reduced in patients with active CD, but there is a potential for the quantification of motility by implementing the MI as an adjunctive parameter in MRE assessment. Further studies are required to determine whether normalisation of the MI may be a useful treatment target in patients with CD.

### 2.8. Fibrofatty Proliferation/Creeping Fat

Fibrofatty proliferation presents as fat wrapping around an inflamed bowel loop. On MRE, this presents as reduced signal intensity on T1-weighted images (see Figure 3) [11]. The presence of fibrofatty proliferation is pathognomonic for CD and is suggestive of longstanding disease [38]. Studies have demonstrated that creeping fat may indeed persist despite endoscopic response to therapy [17,39]. In a prospective study of 28 patients receiving anti-TNF therapy for CD, creeping fat did not significantly differ between baseline and repeat MRE at 52 weeks [39]. Moreover, the presence of creeping fat was not associated with disease relapse [39]. Therefore, it may be assumed that the presence of creeping fat is an established MRE parameter that is not affected by endoscopic or mucosal disease activity and is suggestive of chronic CD. Fibrofatty proliferation on MRE has a fair inter-observer reliability, with one study demonstrating an interclass correlation coefficient of 0.47 [21].

### 2.9. Mesenteric Vascularity (Comb Sign)

The comb sign is an MRE parameter associated with increased CD activity and occurs as a result of vascular engorgement of the vasa recta that extend perpendicularly from the mesentery to the inflamed bowel wall. The comb sign often occurs in parallel with increased stranding of mesenteric fat, and in conjunction, they are known to correlate with extensive and active CD [40].

### 2.10. Inflammatory Lymph Nodes

Inflammatory lymph nodes also often occur in conjunction with increased mesenteric vascularity. Measurements of >15 mm in the short axis are considered pathological and may be indicative of active CD [12]. The presence of pathological lymph nodes may be significant in differentiating between endoscopic activity and remission. Rimola et al. demonstrated that CD segments associated with active endoscopic disease were more likely to demonstrate pathological regional lymphadenopathy in the adjacent mesentery [18]. Though more likely to be associated with endoscopically active CD, as an individual parameter, the presence of lymph nodes was poorly sensitive, with only 17% of non-ulcerated inflamed bowel segments having associated pathological lymphadenopathy [18]. When compared with histopathological resection specimens, Zappa et al. were unable to demonstrate the ability of lymph nodes to distinguish between mild and severe grades of inflammation [13]. Thus, in the absence of other parameters of disease, the presence of pathological lymph nodes cannot be utilised to denote the likelihood of endoscopically active disease via MRE.

### 2.11. Abscess Detection

Abscesses are a known complication of advanced penetrating CD. On MRE, they are characterised by rim enhancement on post-contrast T1-weighted images and a high central signal on T2-weighted sequences [11]. Abscesses are frequently surrounded by fat stranding. In a previous meta-analysis, the pooled sensitivity of MRI for the detection of abscesses was 86%, with a specificity of 93% when compared to histopathology as the reference standard [41].

### 2.12. Enteric Fistula

Enteric fistulas are almost always seen proximal to a region of bowel stenosis. On cross-sectional imaging, enteric fistulas can be seen as a connection between bowel segments, intrabdominal organs, and/or skin. Fistula tracts are best appreciated on T1-weighted post-contrast sequences and demonstrate tract wall enhancement. In a meta-analysis, the pooled sensitivity of MRE for the diagnosis of fistulas was 76%, and the specificity was 96% when compared to histopathology specimens as the reference standard [41].

## 3. Overall Accuracy of MRE for Detection of SMALL Bowel CD Activity

The accuracy of MRE in small bowel CD has been extensively studied against multiple reference standards, including clinical symptoms and biochemistry, endoscopic activity, histological activity, and surgical resection specimens. In a recently published large multicentre study, the sensitivity and specificity of MRE in detecting small bowel CD was 97% and 96% compared to a panel-derived reference including ulceration as seen at endoscopy, a measured CRP concentration of >8 mg/L, a measured calprotectin concentration of >250 μg/g, histopathological evidence of acute inflammation based on a biopsy sample, or surgery within 2 months of MRE [42].

Similarly, the meta-analysis by Qiu et al. reported a pooled sensitivity and specificity for the detection of active small bowel CD with MRE of 88% and 81%, respectively, using ileocolonoscopy or surgical resection specimens as the reference [43]. MRE also demonstrated an excellent capacity at discriminating between active and inactive disease (area under the curve (AUC) 0.91) [43]. No meta-analysis has assessed the accuracy of MRE or DWI-MRE in isolation versus ileocolonoscopy alone as the reference standard in the detection of active versus inactive small bowel CD. Finally, the method of oral contrast delivery appears to have minimal impact in the detection of small bowel CD, and there is no observed difference in the accuracy elicited between MRE and MR enteroclysis [44].

## 4. The Utility of MRE Indices for Evaluation of CD Activity at MUCOSAL Level

MRE activity indices facilitate the standardisation of MRE reporting in CD and have thus received increasing attention. Through the incorporation and validation of multiple parameters, MRE indices enable reproducibility and responsiveness, which provide the opportunity to confidently apply MRE to assess CD activity objectively, monitor the response to therapy, and evaluate outcomes in clinical trials involving MRE. Currently developed indices have proven useful in the assessment of transmural disease, yet comparatively little is known of the utility of MRE indices in superficial mucosal disease [45]. MRE indices for CD activity developed against ileocolonoscopy or histopathological specimens are summarised in Table 2, with cut offs for indices-based CD activity summarised in Table 3.

The first developed CD activity index was the magnetic resonance index of activity (MaRIA) with validation performed against ileocolonoscopy as the reference standard. The index was developed though the segmental bowel assessment of MRE assessment parameters including BWT, relative contrast enhancement (RCE), the presence of oedema, and ulceration, with a global MaRIA score being the sum of six bowel segments (terminal ileum, ascending colon, transverse colon, descending colon, sigmoid colon, and rectum) scores. Within the developmental study, the segmental MaRIA demonstrated a strong correlation with the corresponding bowel segment on endoscopy in both the large and small bowel (*r* = 0.82, *p* < 0.001) [10]. Moreover, the MaRIA index demonstrated an excellent capacity for the detection of CD activity in both the large and small bowel (sensitivity of 81%, specificity of 89% (AUC = 0.89)) [10]. In a subsequent validation study, a segmental MaRIA score ≥7 demonstrated a sensitivity of 87% and a specificity of 87% (AUC 0.93) for active endoscopic disease, with a MaRIA score ≥11 demonstrating a sensitivity of 92% and a specificity of 92% (AUC 0.96) for severe endoscopic disease [18]. The MaRIA score was also found to be a reliable index in assessing the response to therapy and endoscopic mucosal healing [17].

To overcome the time-consuming and complex measurements in the original MaRIA score, a simplified MaRIA (sMaRIA) was developed utilising non-intravenous contrast-dependent MRE parameters such as BWT, mural oedema, fat stranding, and mucosal ulcerations [19]. In the developmental study, an excellent correlation between the sMaRIA and endoscopic activity with the large and small bowel (*r* = 0.83, *p* < 0.001) was observed [19]. A sMARIA score >1 identified a segment of active endoscopic CD with 90% sensitivity and 81% specificity (AUC 0.91), with a score >2 detecting severe endoscopic lesions with 85% sensitivity and 92% specificity (AUC 0.94) [19]. Within the validation cohort, the sMaRIA demonstrated an excellent capacity to assess the response to therapy, with a significant reduction in segmental scores commensurate with the achievement of endoscopic mucosal healing (sMaRIA baseline 3.30 ± 1.90 vs. sMARIA post-treatment 0.75 ± 1.35; *p* < 0.001) [19]. Apart from maintaining performance in the absence of intravenous contrast, the sMaRIA also boasted significantly shorter reporting times compared to the original MaRIA, with a reduction in the mean calculation time (4.77 min vs. 17.14 min; *p* < 0.001) [16,19]. In a recent study of patients with established small bowel CD, the sMaRIA was validated against terminal ileal histology obtained through ileocolonoscopy as a reference standard, demonstrating a sensitivity and a specificity for histologically active disease of 83% and 41%, respectively [46]. Though the sensitivity was preserved compared to internal validation studies, the striking reduction in specificity is of particular concern, given this may indicate a decreased ability of MRE to rule out active mucosal disease.

The Crohn’s Disease MRI Index (CDMI) and the London Index are the only activity indices developed against surgical resection specimens as a reference standard [26]. Incorporating a T2 mural score and BWT, the CDMI demonstrated a positive correlation with histology (τ = 0.40 (0.11–0.64), *p* = 0.02) [26]. Furthermore, a CDMI score >4.1 demonstrated a sensitivity of 81% and a specificity of 70% (AUC 0.77) for the detection of active CD within the terminal ileum [26]. When compared to disease activity assessment graded through colonoscopy, the CDMI had only a moderate correlation (*r* = 0.59 (0.47–0.69)) [47]. Developed from the same study as the CDMI, the London Index is a qualitative index incorporating mural thickness, mural T2 score, perimural T2 signal, and contrast enhancement as MRE parameters [26]. The London Index has demonstrated a sensitivity of 87% and a specificity of 70% (AUC 0.83, applying >3 as a cut-off point) [26]. The CDMI and London indices have also been directly assessed against terminal ileal histology using pre-established cut-offs of >4.1 and >3 [46]. When discriminating between histologically active and inactive disease, the CDMI demonstrated a sensitivity and specificity of 76% and 64%, respectively [46]. The London Index had a slightly higher sensitivity of 81% but a lower specificity of 41% [46].

The Clermont score is a DWI-based index for CD activity. It was uniquely developed against the MaRIA as the reference standard rather than histology and ileocolonoscopy [48]. Incorporating DWI through measurement of the apparent diffusion coefficient (ADC), BWT, and/or presence of ulcers and oedema, the Clermont score demonstrated a high correlation (rho = 0.99) with the MaRIA score in ileal CD, albeit not in colonic CD (rho <0.80) [49]. A Clermont score of >8.4 was found to be predictive of active ileal CD (AUC of 0.99), with a value of ≥12.5 being highly predictive of severe ileal CD (AUC = 0.99,) [49]. When assessed against active endoscopic disease within the terminal ileum, the Clermont score showed a sensitivity and specificity of 85% and 64%, respectively [50]. When compared to active ileal histology, there was a marginal reduction in sensitivity and specificity of 82% and 55%, respectively [50]. Another DWI-based score known as the Nancy Index was developed and validated in CD with colonic involvement only [51]. A subsequent study of 20 patients with ileocolonic CD found a Nancy score <2 demonstrating an AUC of 0.80 for discriminating mucosal healing, using ileocolonoscopy as the reference standard [52]. Despite this, the Nancy Index has not been applied widely as a MRE index of activity and requires further study assessing its accuracy with reference to histology in CD.

Another MRE activity index in CD, known as the five-point magnetic resonance enterocolongraphy (MREC) score, has been validated against the ileocolonoscopic activity in small and large bowel CD. The five-point score showed a moderate correlation with the CDEIS (r = 0.51–0.66 between observers) [53]. MRE has also been utilised as part of assessing overall bowel damage in patients with CD through measurement of the Lemann Index [54]. The Lemann Index requires multiple inputs, including clinical, endoscopic, and multiple imaging modalities, and has been discussed elsewhere in the literature [54].

The MaRIA, sMaRIA, CDMI, London Index, and Clermont Index have all demonstrated good inter-rater reliability via external validation studies, with wall thickness being a common parameter amongst these indices [16,50]. At a mucosal and histological level, all indices appear to be highly sensitive in detecting endoscopically and histologically active disease, though all indices tended to yield reduced specificity, particularly when ileal histology is used as the reference standard [46,55]. The lower specificity may reflect the lack of an agreed upon and/or universally applied histopathological activity scoring system in CD and potentially the high inter-observer variability of histological assessment in CD. Another plausible explanation is the variable quality of acquired images that may impede accurate MRE assessment. The incorporation of quality metrics within MRE is poorly described within the literature thus far, despite the inherent importance of parameters such as bowel distension and motion artifacts for optimal image assessment. Within the currently developed indices, none appear to have a clear diagnostic advantage compared to the others at the mucosal level. Specifically, within the development study, there was no difference in diagnostic performance between the sMaRIA and original MaRIA indices for the detection of active CD compared to ileocolonoscopy as the reference standard [19]. However, compared to the Clermont and CDMI, the original MaRIA was superior at discriminating between active and inactive endoscopic disease, though no significant difference was observed in the detection of CD-related ulcers with ileocolonoscopy [55].

**Table 2 diagnostics-13-01061-t002:** Summary of cross-sectional modality indices of activity in CD.

Study	Index/Modality	Study Design	*N* =	Reference Index	Parameters Included	Segment Assessed	Formula	Statistical Assessment	Bowel Prep./Rectal Water Instilled?	External Validation?
Rimola et al. [10]	MaRIA/MRE	R	50	C (CDEIS)	BWT RCE Oedema Ulceration	TI AC TC DC SC Rectum	MaRIA (segmental) = 1.5 × BWT (mm) + 0.02 × RCE + 5 × oedema + 10 × ulceration MaRIA (global) = sum of all segments	Correlation with CDEIS: *r =* 0.82 Detection of disease activity (ileum and large bowel): AUC = 0.891, sensitivity = 0.81, specificity 0.89	Yes/Yes	Yes [47,56]
Steward et al. [26]	CDMI/MRE	R	16	Histology (AIS)	BWT Mural T2 score	TI	CDMI = 1.79 + 1.34 mural thickness + 0.94 mural T2 score	Correlation with histology (AIS): Kendalls tau = 0.40, 95% CI (0.11–0.64) Detection of active TI disease using cut off >4.1: AUC 0.77, sensitivity 81% (95% CI (54–96)), specificity 70% (35–93)	No/No	Yes [47]
Steward et al. [26]	London Index/MRE	R	12	Histology (AIS)	BWT Mural T2 score Perimural T2 signal Contrast enhancement	TI	London Index = mural thickness + mural T2 score + perimural T2 signal + contrast enhancement	Detection of active TI disease using cut off >3: AUC 0.83, sensitivity 87% (61–98), specificity 70% (35–93)	No/No	Yes [46]
Buisson et al. [48]	Clermont/MRE	P	31	MRE (MaRIA)	BWT Ulcers Oedema ADC	TI	Clermont = −1.321 × ADC (mm^2^/s) + 1.646 × wall thickening + 8.306 × ulcers + 5.613 × oedema + 5.039	No difference found between original MaRIA score (*R*² = 0.998) and Clermont score (*R*² = 0.989)	No/No	Yes [56]
Ordas et al. [19]	sMaRIA/MRE	R	98D 37V	C (CDEIS)	BWT oedema Perienteric fat stranding Ulcers	TI AC TC DC SC Rectum	sMaRIA (segmental) = 1 (1 × thickness >3 mm) + (1 × oedema) + (1 × fat stranding) + (2 × ulcers) sMaRIA (global) = addition of all segments	Correlation with CDEIS: *r =* 0.83 Detection of disease activity (ileum and large bowel): AUC = 0.91, sensitivity 90%, specificity 81%	Yes/Yes	Yes [57]
Thierry et al.	Nancy/MRE	P	20	C (CDEIS)	Ulceration Parietal oedema BWT differentiation between (sub) mucosa and muscularis propria Rapid contrast enhancement, DWI hyperintensity	TIRCTCLCSCRectum	Nancy Index (segmental) = ulceration + parietal oedema + BWT + differentiation between (sub)mucosa and muscularis propria + rapid contrast enhancement + DWI hyperintensity 1 point for each parameter present. Maximum 6 points.	Detection of disease activity: AUC 0.80, sensitivity 92%, specificity 68%	No/No	Yes [52]

MaRIA: Magnetic Resonance Index of Activity; C: colonoscopy; CDEIS: Crohn’s Disease Endoscopic Index of Severity; BWT: bowel wall thickness; RCE: relative contrast enhancement; TI: terminal ileum; AC: ascending colon; RC: right colon; LC: left colon; TC: transverse colon; DC: descending colon; SC: sigmoid colon; R: retrospective; AIS: acute inflammation score; P: prospective; ADC: apparent diffusion coefficient; MEGS: Magnetic Resonance Enterography Global Score; CRP: C-reactive protein; HBI: Harvey Bradshaw Index; FCP: faecal calprotectin; LN: lymph nodes; C: caecum; J: jejunum; I: ileum; sMaRIA: Simplified Magnetic Resonance Index of Activity.

**Table 3 diagnostics-13-01061-t003:** Cut-off values for active and severe disease with different MRE indices.

Score	Active Disease	Severe Disease
MaRIA (segmental)	≥7	≥11
sMaRIA (segmental)	>1	>2
Clermont (segmental)	>8.4	≥12.5
Nancy (segmental)	≥2	
CDMI	≥4.1	
London Index	>3	

## 5. The Utility of MRE to Assess Response to Therapy at Mucosal Level Either via Individual MRE Parameters or MRE Indices

Studies have purported the reliability of MRE as a surrogate marker of the response to therapy in CD, though critically, expert consensus definitions of the change in MRE parameters that defines a clinically meaningful response are yet to be elucidated [17,58,59,60]. As a result, there is significant heterogeneity in the grading of MRE parameters in studies assessing the response to therapy in CD.

MRE parameter response in CD has been demonstrated as early as 12 weeks after the initiation of CD therapy [17]. In a prospective study of 48 patients with CD undergoing induction therapy with corticosteroids or anti-TNF agents, MRE predicted endoscopic ulcer healing with a sensitivity and specificity of 75% and 80%, respectively, at 12 weeks, compared to endoscopy as the reference standard. In patients that demonstrated endoscopic ulcer healing at 12 weeks, MRE parameters such as BWT, presence of intramural oedema, ulceration, presence of lymph nodes, degree of contrast enhancement, perienteric vascularisation, and fat stranding improved and/or worsened significantly compared to baseline (each *p* < 0.05) [17]. Importantly, amongst the endoscopic segments without ulcer healing, none of the MRE parameters demonstrated a significant change from baseline measurements [17]. Only fibrofatty proliferation remained unchanged in treatment responders, which is consistent with other studies and likely represents chronic damage seen with CD, hence being less or not amenable to change [17,39]. Furthermore, the presence of fibrofatty proliferation within ileal segments on pre-treatment MRE has been associated with an increased likelihood of treatment failure in CD [61].

Similarly, despite a longer follow-up period of 46 weeks post initiation of CD therapy, BWT, the presence of intramural oedema, and ulceration each showed a significant reduction in segments with endoscopic ulcer healing compared to those without [16]. Further, in a recent study of 24 patients with ileal CD assessed through tandem ileocolonoscopy and MRE up to 96 weeks, the mean length of disease within the terminal ileum was significantly reduced in endoscopic responders compared to non-responders [62]. Thus, the change in the length of the diseased segment from the baseline may be another MRE parameter indicative of the endoscopic response to therapy in CD.

The inter-rater reliability of individual parameters of MRE pre and post treatment in CD has been studied extensively with varied outcomes [47,63]. Higher rates of inter-rater reliability within MRE parameters in CD have been demonstrated when abdominal focused radiologists assessed and reported the respective MREs [21,58]. In a recent study by Tsai et al. in which non-abdominal-focused radiologists performed MRE assessments, pre-treatment BWT and the presence of diffusion restriction demonstrated a moderate inter-rater agreement (k/ICC = 0.43–0.56) with the type and degree of contrast enhancement, but only a fair agreement (k/ICC = 0.22–0.40) with the degree of intramural oedema [63]. In post-treatment MREs, BWT demonstrated a substantial agreement (ICC = 0.69) with the type of contrast enhancement, and the presence of diffusion restriction demonstrated a moderate agreement (k/ICC = 0.45–0.48) [63]. Intramural oedema again showed only a fair inter-rater agreement (k = 0.30), with the degree of contrast enhancement having only a slight inter-rater agreement (k = 0.10) [63]. There was no correlation between pre- and post-treatment assessments of mucosal ulceration, a key parameter in MRE indices (k(pre-treatment) = −0.071, k(post treatment) = −0.042). Hence, there is an apparent net gain in accuracy when abdominal-focused radiologists perform assessment(s) of MRE parameters in CD. Furthermore, based on currently available evidence, MRE parameters that are abnormal prior to the initiation of therapy appear to be reliable markers of disease progression and the response to therapy at the mucosal level subsequently.

The utilisation of a combination of parameters via one of the published MRE indices for monitoring the response to therapy in CD is an attractive proposition. However, MRE indices have yet to be widely applied beyond clinical studies and/or trials.

The MaRIA score has a demonstrated capacity for detecting the response to therapy at the mucosal level [17,19]. In CD, in patients who demonstrated endoscopic ulcer healing at 12 weeks after the initiation of therapy, a significant reduction in a segmental MaRIA score was observed [17]. Conversely, in segments without ulcer healing, no change in MaRIA was elicited at week 12. Furthermore, a segmental MaRIA score of <11 demonstrated a sensitivity, specificity, positive predictive value, and negative predictive value of 94%, 69%, 94%, and 67%, respectively, for predicting endoscopic ulcer healing within the corresponding endoscopic segment [17]. Similarly, a segmental MaRIA score of <7 demonstrated a sensitivity, specificity, positive predictive value, and negative predictive value of 85%, 78%, 92%, and 63%, respectively, in predicting mucosal healing within the corresponding endoscopic segment [17].

Similarly, the sMaRIA also appears to be useful for detecting the response at the mucosal level [16,19]. In a prospective study of patients with CD using ileocolonoscopy as a reference standard, a sMARIA of <2 at week 46 had a sensitivity, specificity, positive predictive value, negative predictive value, and overall accuracy of 90%, 88%, 94%, 78%, and 89%, respectively, for identifying ulcer healing. Yet in segments with persistent endoscopic ulceration, the sMaRIA did not elicit a significant change following 46 weeks of CD therapy compared to baseline [16].

The utility of the CDMI in measuring the response to anti-TNF therapy was evaluated in a retrospective cohort of 50 CD patients against a reference of a global physician’s assessment comprising a composite of clinical, endoscopic, stool, and serum biomarker assessment performed within 1 month of the MRE [58]. The physician’s assessment categorised patients as anti-TNF responders or non-responders. In anti-TNF responders, the mean CDMI decreased from 5.19 to 3.12 (*p* < 0.0001), but no significant change was seen in anti-TNF non-responders [58]. A key limitation within this study is the lack of endoscopic correlation for the entire cohort, given the reference standard was a composite of multiple parameters. Thus, the CDMI scored should be used with caution until further studies have assessed its capacity in determining the response to therapy in CD.

The Clermont score has also been validated as a reliable index in detecting mucosal healing, though its capacity for detecting the response to therapy has been mostly limited to studies using clinical assessments as the reference standard [64,65]. In a small retrospective study of 24 patients receiving anti-TNF therapy for CD, there was a significant change in mean pre- and post-treatment ileal Clermont scores between endoscopic anti-TNF responders versus non-responders [62]. Pre- and post-treatment Clermont scores demonstrated an accuracy of 73.9% and 91.3% compared with the corresponding ileal segment assessed via ileocolonoscopy [62]. The Nancy score was assessed for responsiveness to change in a small cohort of 20 patients after the initiation of a biologic agent [52]. In the 11 CD patients who achieved endoscopic mucosal healing, a significant decrease between baseline and post-treatment Nancy score was observed, and importantly, no change was observed in the non-responder cohort [52].

Based on the current literature, MRE indices of activity show promise in the assessment of the response to therapy. The sMaRIA has multiple practical advantages over the original MaRIA, including a shorter calculation time, plus it precludes the need for intravenous contrast. Studies assessing inter- and intra-rater reliability of these indices are limited, particularly in the setting of interpretation by non-abdominal-focused radiologists. Hence, based on the quality of data available in relation to assessing the response to therapy at the mucosal level, the MaRIA and sMaRIA should currently be favoured over other MRE indices of activity in CD.

## 6. The Utility of MRE for Detection of Post-Operative Recurrence in CD

Post-operative recurrence of CD has been reported in up to 90% of patients at one year post resection [66,67]. Though MRE is an accepted non-invasive imaging modality for the assessment of post-operative CD recurrence, ileocolonoscopy at 6 months remains the gold standard [68]. Despite being widely performed, few studies have specifically assessed the accuracy of MRE and/or MR enteroclysis in the assessment of post-operative CD recurrence. Sailer et al. prospectively evaluated 30 patients utilising MR enteroclysis for a suspected recurrence following intestinal resection and found a moderate correlation (kappa 0.67, intra-observer variability 77.8%) with endoscopy (Rutgeerts score) as the reference standard [69]. MR enteroclysis was able to distinguish mild endoscopic recurrence (Rutgeerts i1-2) and severe endoscopic recurrence (Rutgeerts > i3) with an excellent intra-rater agreement (kappa 0.84, 95% intra-rater variability). However, the capacity for differentiating mild endoscopic recurrence (Rutgeerts i0 vs. i1-2) was not assessed [69]. In a subsequent study by the same authors, MR enteroclysis was able to detect mild endoscopic recurrence (Rutgeerts i1-2) with 100% sensitivity, though the sample size was limited to five patients; thus, larger prospective studies are required [70].

Moreover, in a recent meta-analysis, the pooled sensitivity and specificity of MRE for the detection of endoscopic recurrence (Rutgeerts > i1) was 97% (95% CI, (89–100%)) and 84% (61–96%), respectively [71]. MRE showed an excellent discrimination capacity between active and inactive endoscopic disease, with an AUC of 0.98 [71]. Despite these impressive results, only three studies with a total of 76 patients were included in the meta-analysis, thereby limiting the applicability of these data, particularly in mild post-operative CD recurrence (Rutgeerts i1-2). More recently, a large study of 216 CD patients with post-ileocaecal resection who underwent tandem ileocolonoscopy and cross-sectional imaging (MRE or CTE) found BWT, luminal narrowing, mural hyper-enhancement, and length of the disease on imaging to be significantly associated with endoscopic recurrence (each *p* < 0.05) [72]. Critically, when defining endoscopic recurrence as Rutgeerts ≥ i2a, using cross-sectional imaging to differentiate active endoscopic disease demonstrated a high sensitivity of 89%, yet a low specificity of merely 32%, thus limiting its role to screening for disease recurrence with confirmation required through ileocolonoscopic examination [72].

A Magnetic Resonance Imaging Index to Predict Crohn’s Disease Postoperative Recurrence (MONITOR) Index has been developed and internally validated [73]. Within this study, DWI signal increase, increased BWT, length of the disease segment of >20 mm, the presence of contrast enhancement, T2 signal increase, oedema, and ulcers at the anastomotic site and terminal ileum were each independently associated with endoscopic disease recurrence (Rutgeerts > i2), with the presence of ulcers on MRE demonstrating the strongest association for recurrence. The MONITOR index consists of a sum of the seven parameters independently associated with active endoscopic disease, with higher weighting given to the presence of ulcers. The index demonstrated an AUC of 0.85 (0.73–0.97) for differentiating active and inactive disease compared to the reference standard of active endoscopic CD (Rutgeerts > i2). Furthermore, in the validation cohort, a MONITOR index of ≥1 gave a sensitivity and specificity of 87% and 75%, respectively [73]. Given the small validation cohort (*n* = 17), external validation of the MONITOR index is awaited prior to widespread clinical application. The index is, however, a logical step in standardising MRE reporting for both the post-operative setting and, more widely, in the assessment of CD activity.

## 7. Persistently Abnormal MRE Findings in the Setting of Normal Endoscopy

Given that inflammation in CD may extend transmurally, the scenario of persistently abnormal MRE findings but a demonstration of mucosal healing via endoscopy can be encountered [74,75,76]. Broadly, this dichotomy may represent the juxtaposition of active and fibrostenosing CD. Moreover, intuitively, the pattern of progression of mural inflammation is variable, and the disease process may begin and/or predominate at the mucosal level and extend to the mesentery or vice versa. Similarly, the process of healing could occur variably in either manner. Hence, a mucosal to mesenteric healing pattern may reflect a scenario in which ileocolonoscopy findings are normal, yet abnormal MRE parameters such as BWT persist.

The prognostic implications of persistent MRE disease in the absence of mucosal inflammation at colonoscopy have been evaluated in a recent retrospective study of 112 patients with known CD who had a normal ileoscopy, but who had findings of active disease on cross-sectional imaging (i.e., CTE and/or MRE) performed within 30 days of each other [76]. Of the 88 patients who had both a negative ileocolonosccopy and ileal biopsies, 44 (50%) had moderate/severe inflammation on MRE or CTE, with 45%, 32%, and 11% having proximal small bowel inflammation, demonstrable strictures, or fistulas on imaging, respectively [76]. Subsequently, 59 (67%) patients had documented progression of known small bowel CD, which was defined as a requirement for subsequent surgical resection, radiological worsening on future MRE/CTE, and/or ulcers at subsequent ileoscopy [76]. Thirteen (15%) of these patients were deemed to have achieved a radiologic response based on subsequent MRE/CTE, and 16 (18%) had unchanged findings on subsequent scanning [76]. Of the 24 patients who had active CD on ileal histology yet no inflammatory activity on endoscopy, 14 (63%) had a progression of small bowel CD, with no patients demonstrating a radiologic response, and the remaining 9 (37%) had unchanged findings on subsequent MRE/CTE [76]. Importantly, this highlights the transmural component of CD, with active disease occurring beyond the mucosal surface of the lumen, which potentially may provide false reassurance in those with normal mucosal findings endoscopically. Though not always achievable with current medical therapy, treatment targets beyond MH should therefore be contemplated.

Transmural healing (TH) has been proposed as a treatment target in CD [7,8,54]. A recent systematic review highlighted the heterogeneity of definitions of TH across multiple modalities of bowel imaging that have incorporated parameters, such as varying grades of BWT, contrast enhancement, and complications [74]. Though clearer definitions of TH in MRE are needed, early studies have shown that the achievement of TH is associated with a significant reduction in the requirement for surgical intervention, hospitalisation, corticosteroids, and/or therapeutic changes/escalation [75,77,78,79]. Thus, despite the achievements of MH, cross-sectional evaluation may provide additional prognostic implications, and hence consideration should be given to performing both endoscopy and imaging when evaluating treatment response in patients with small bowel CD.

## 8. The Impact of Concurrent Fibrosis on the Utility of MRE in the Assessment of Mucosal Activity in CD

Despite advances in anti-inflammatory therapies for CD, there have been limited developments addressing the prevention and/or reversal of intestinal fibrosis [80]. Population studies have suggested that up to 30% of patients with CD develop fibrostenosing disease within 20 years of diagnosis [81,82]. The development of fibrosis typically results in gradual luminal narrowing and stricture formation [83]. Evaluating the phenotype (i.e., inflammatory versus fibrostenosing) of small bowel strictures in CD has prognostic importance and may lead to different treatment algorithms [84], though strictures often have a mixed inflammatory and fibrotic component [13].

MRE and its various techniques have been studied in relation to characterising phenotypes of CD strictures, though the exact composition of strictures is often impossible to determine based on radiological findings alone [13,14,25,85,86]. In fibrotic CD, MRE parameters can display similar abnormalities as those seen in active inflammatory CD, thus posing a challenge to determining the nature of the disease, particularly where there is no or minimal endoscopic activity.

Zappa et al. retrospectively evaluated 53 patients who underwent MRE within 3 months of surgical resection and graded histopathological specimens with an internally developed fibrosis score. MRE parameters comprising increased BWT on T2 and T1 sequences, increased T2 mural signal intensity, fibrofatty proliferation, and the presence of a fistula were each associated with advancing histopathological grades of fibrosis [13]. However, the same MRE parameters were similarly associated with histopathological grades of inflammation, hence limiting their individual ability to discriminate between fibrotic and inflammatory CD.

Rimola et al. retrospectively studied aspects of intravenous contrast parameters to discriminate between CD-related fibrosis and inflammation with MRE. Forty-one patients had an MRE within 120 days prior to a planned small intestinal resection, with histopathological grades of fibrosis found to be significantly associated with the percentage of the enhancement gain, the pattern of enhancement, and the presence of stenosis [14]. Using the percentage of the enhancement gain, MRE appeared able to separate grades of fibrosis (AUC 0.93, sensitivity 94%, specificity 89%), but it could not differentiate between fibrosis and inflammatory disease [14].

DWI has also been examined as a potential MRE parameter for discriminating between fibrosis and active inflammatory CD, yet it has gleaned mixed results [33,86]. Wagner et al. applied pre-operative DWI-MRE to a cohort of 35 CD patients undergoing ileal resection; however, DWI sequences were unable to discriminate between fibrosis and inflammatory activity detected in surgical resection samples [86]. Lower ADC values were observed with both advancing grades of inflammation and fibrosis [25,86].

Dynamic contrast enhancement (DCE)-MRE quantitatively measures perfusion by measuring the contrast signal within the bowel wall. It has been investigated as a tool to determine CD activity and to discriminate between fibrosis and active inflammatory disease [25,33]. Tielbeek et al. found the maximum contrast enhancement and the slope of increase after contrast injection had a significant correlation with differences between grades of fibrosis on histopathological resection samples (*p* < 0.05 for all) [25].

Other promising modalities in differentiating fibrosis and inflammatory CD include magnetisation transfer (MT)-MRE and positron emission tomography (PET) [87,88]. Image contrast enhancement in MT-MRE is representative of the fraction of macromolecules, such as collagens, in intestinal tissue. Despite initial studies showing a positive correlation between MT-MRE and histopathologic fibrosis, more recent studies have demonstrated a weak correlation (r = 0.28) [15,87]. Using surgical histopathological resection specimens as a reference standard, PET-MRE demonstrated an acceptable discrimination (AUC 0.77 (0.56–0.91)) in differentiating inflammatory and fibrotic lesions [88,89]). Though promising, PET-MRE increases radiation exposure to patients and is associated with significant costs and resource limitations, thus potentially limiting its widespread application.

Despite the numerous studies, ongoing uncertainties of MRE findings in correctly predicting histopathological fibrosis and stricture composition limit its use in discriminating between both phenotypes [33]. Moreover, histopathological analysis of resected bowel in CD consistently suggests the dual presence of inflammatory and fibrotic tissue [13,14]. Abnormal MRE parameters can pose significant challenges to interpretation, particularly in the setting of a normal ileocolonoscopy. It is anticipated that further exploration of different sequences and techniques within MRE might facilitate improved differentiation of fibrotic and inflammatory phenotypes of CD.

## 9. Limitations of MRE

Though MRE is commonly used in the assessment of small bowel CD, limitations of the procedure include the duration of the examination, the requirement of high-volume oral and IV contrast, along with cost, resource, and access issues in many healthcare jurisdictions. In addition, a potential for misdiagnosis through failure to identify active CD when present on ileocolonoscopy or ‘over-call’ of MRE findings suggestive of active disease remains a clear possibility. In such scenarios, optimisation or MRE techniques (e.g., the volume and timing of oral and IV contrast, the use of anti spasmodics) could improve the possibility of more accurate disease activity detection [90]. Furthermore, alternative imaging modalities such as IUS and CTE could be of particular use as a second method to confirm or more definitively exclude unclear findings [91,92].

This review highlights the challenges of assessing CD activity at the mucosal level, particularly in milder and/or partially treated, non-ulcerative disease phenotypes that appear to be problematic for MRE to interrogate accurately. In these circumstances, ileocolonoscopy is the preferable and likely more accurate assessment modality. Moreover, in relation to disease distributed proximal to the terminal ileum, video capsule endoscopy has been shown to be an accurate modality for CD activity assessment and should be considered when both ileocolonoscopy and MRE are unable to detect suspected disease [3,93]. A potential strategy of assessing disease activity in small bowel CD is illustrated in Figure 5.

## 10. Conclusions

MRE in small bowel CD is an established modality of CD activity assessment. With the current Selecting Therapeutic Targets in Inflammatory Bowel Disease (STRIDE) guidelines recommending assessments every 6–12 months in patients with CD [7], accurate options for the assessment of CD activity beyond ileocolonoscopy are needed. Significant advances in the technical performance and interpretation of MRE have recently been published and have led to the enhanced accuracy, and therefore utility, of MRE in CD activity assessment. Novel techniques in MRE have been continually developed, with further developments particularly in artificial intelligence likely to feature in future research, and with incorporation into clinical practice in the coming years. With these technological, procedural, and evidence-driven refinements and advances, MRE is likely to become increasingly central to non-invasive CD activity assessment in routine clinical practice into the future. Nevertheless, challenges currently exist in MRE’s ability to accurately evaluate mucosal disease, particularly in milder or mostly superficial luminal disease. In such settings, alternative modalities of assessment, such as intestinal ultrasound and/or capsule endoscopy, should be sought first to ensure optimal assessment and clinical decision making. Hence, in appropriately selected patients and clinical scenarios, although we may not quite be there, with further high-quality studies exploring MRE in this field, it is certainly foreseeable that MRE could supplant ileocolonoscopy as the ‘go-to’ modality for the assessment of small bowel CD in the future.

## Figures and Tables

**Figure 1 diagnostics-13-01061-f001:**
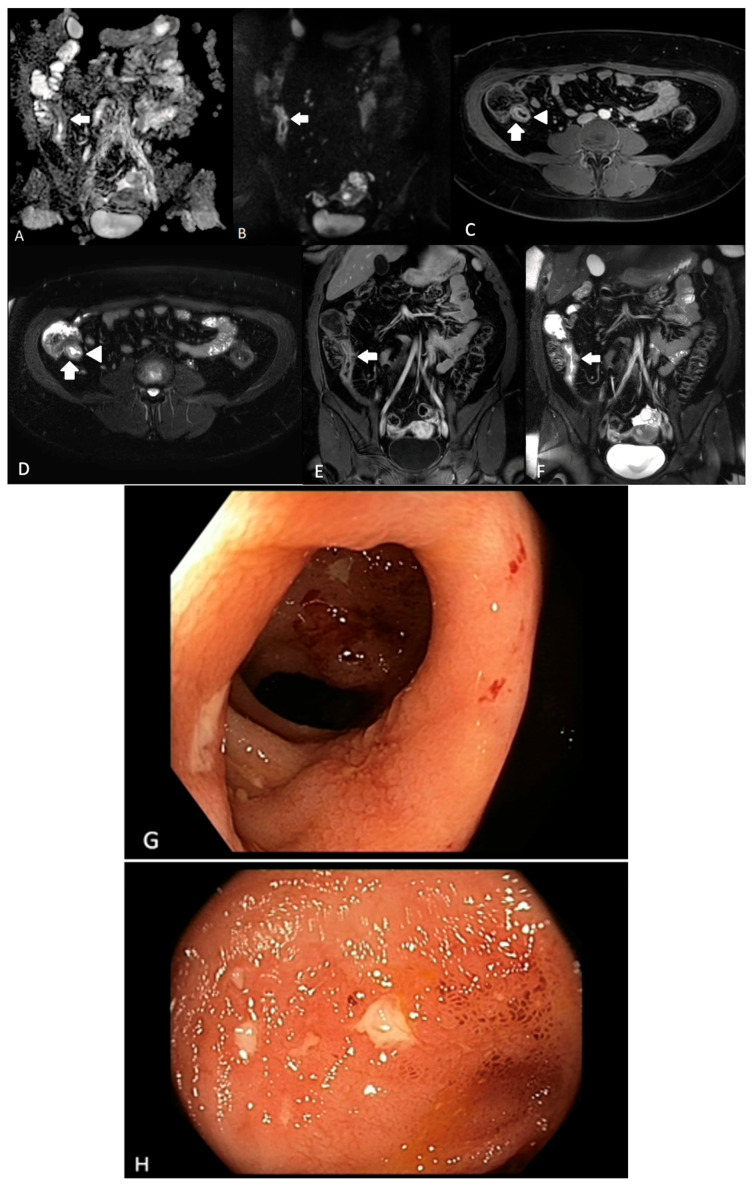
A 31-year-old female with known ileal CD. (**A**) ADC map and (**B**) DWI sequences demonstrating terminal ileal wall diffusion restriction indicating active inflammation (arrow); (**C**) axial T2-weighted fat-saturated sequence demonstrating terminal ileal wall thickening and intra mural oedema (arrow) with ulceration medially (arrowhead); (**D**) axial T2-weighted fat-saturated sequence demonstrating terminal ileal wall thickening and intra mural oedema (arrow) with ulceration medially (arrowhead); (**E**) coronal T1-weighted fat-saturated post-contrast sequence demonstrating terminal ileal wall thickening and hyperenhancement (arrow); (**F**) coronal T2-weighted fat-saturated sequence demonstrating terminal ileal wall thickening and intra mural oedema (arrow), (**G**) and (**H**). Corresponding ileal endoscopic image (SES-CD = 5).

**Figure 2 diagnostics-13-01061-f002:**
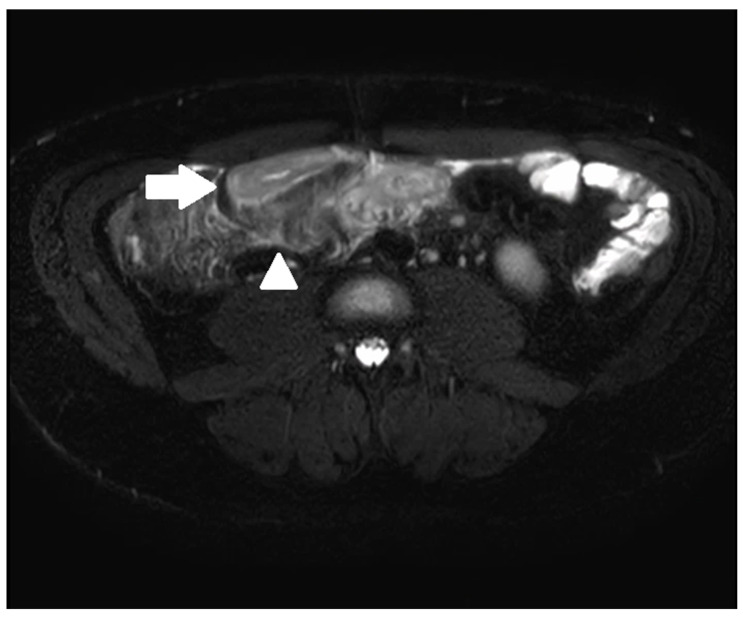
A 33-year-old male with ileal CD. Axial T2-weighted fat-saturated sequence demonstrating terminal ileal wall thickening and intra mural oedema (arrow) with peri enteric oedema (arrowhead).

**Figure 3 diagnostics-13-01061-f003:**
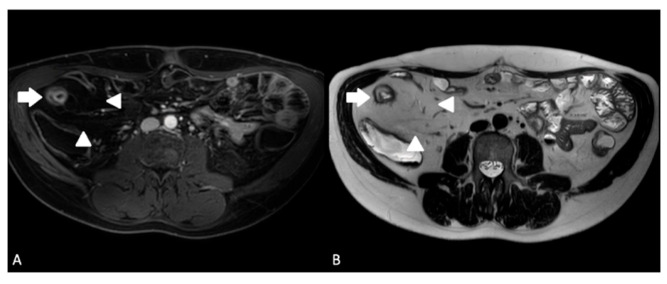
A 24-year-old male with ileal CD. (**A**) Axial T1-weighted fat-saturated post-contrast sequence demonstrating terminal ileal wall thickening and hyperenhancement (arrow) with creeping fat (arrowheads); (**B**) axial T2-weighted sequence demonstrating terminal ileal wall thickening and intra mural oedema (arrow) with creeping fat (arrowheads).

**Figure 4 diagnostics-13-01061-f004:**
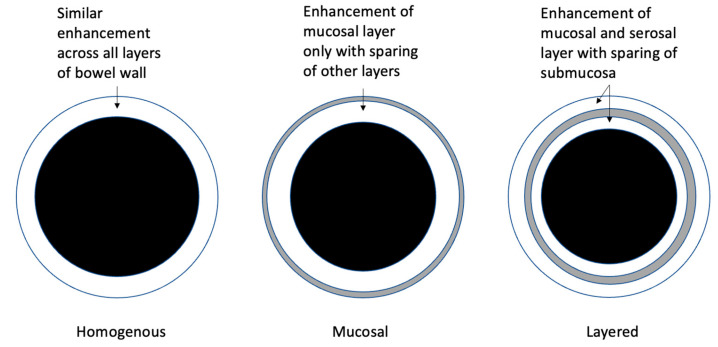
Categories of bowel wall enhancement.

**Figure 5 diagnostics-13-01061-f005:**
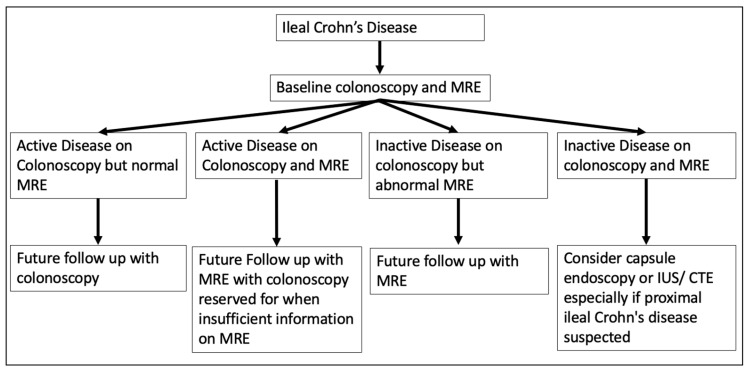
Potential CD activity assessment strategy.

**Table 1 diagnostics-13-01061-t001:** Differentiating MRE parameters between inflammatory and fibrostenosing CD.

Parameter	Active Inflammatory CD	Fibrostenosing CD	Ideal Imaging Sequence to Detect Individual Parameter
BWT > 3 mm	ü	ü	T1- and T2-weighted sequences
Bowel wall contrast enhancement	ü	ü	T1-weighted sequences
T2 mural signal intensity/intramural oedema	ü		T2- and diffusion-weighted sequences
Ulceration	ü		T2-weighted sequences
Increased apparent diffusion coeffienct signal	ü	ü	T2-weighted and T1-weighted post-contrast sequences
Fibrofatty proliferation/creeping	ü	ü	T1- and T2-weighted sequences
Reduced intestinal motility	ü	ü	T1- and T2-weighted sequences

## Data Availability

Not applicable.

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
