# Peer review of "Replacing Endoscopy with Magnetic Resonance Enterography for Mucosal Activity Assessment in Terminal Ileal Crohn’s Disease: Are We There Yet?"

_diagnostics, 2023, doi:10.3390/diagnostics13061061_

Round 1

Reviewer 1 Report

The manuscript discusses the role of MRE in the detection of CD status and brings together a comparison of different techniques and their ability to detect CD status. Though the work is of potential interest, there are several major issues with the presentation. The article should be revised thoroughly before acceptance for improving the presentation style. A few of the specific comments are mentioned below.

1.       Mr enterography should be written properly in the title, using abbreviations in the title should not be done.

2.       The statement “Multiple treatment targets in CD are well established” needs to be revisited as the techniques discussed are used to detect the status of the disease primarily.

3.       Table 1 must use the right sign instead of the cross sign in order to demonstrate clearly the presence under certain parameters.

4.       Figure 4 caption must be elaborated to show different parts of the figure

5.       Some abbreviations must be defined in the first place

6.       What are the chances of misdiagnosis and potential remedies to tackle these? It must be added as a section.

Reviewer 2 Report

This submission reviewed the MR enterography for assessment of terminal or neo-terminal ileum Crohn’s disease.

The title of this review is not appropriate. It is not well reviewed for the assessment of deep part of small intestine, but only of the terminal ileum. There are no description of device-assisted endoscopy.

The authors described as 30% of patients have small bowel involvement. This is too small. Recent papers show more than 70% of patients have small intestinal Crohn’s lesions.

  The aim of this review is to compare MR enterography and endoscopy. But the authors do not refer the papers of the direct comparison of MRE and enteroscopy. They should be referred.

  One of the important limitations of MRE is the less sensitivity of the strictures, that was reported in Gastroenterology in 2014. It should be referred.

  MRE is used as a modality for the bowel damage assessment in Lémann score. It may be mentioned.

  The authors show several scores, but not 5-Point MR Enterocolonography Classification in AJR 2019. It is validated and focused on both colon and small intestine.

 There are small points:

In page12 L426 and L427, the word “inter-rator” are written as “interrator”. They should be should be the same spelling as other sentences .

In Page 14, some words are used italic. They are better to be corrected,

Round 2

Reviewer 2 Report

This submission is well-revised. All comments are answered well.